# False Narratives: Illicit Practices in Colombian Transnational Adoption

**Susan F. Branco** [1,*] and **Veronica Cloonan** [2]

1   Counselor Education, St. Bonaventure University, St. Bonaventure, NY 14778, USA
2   Independent Researcher, Catonsville, MD 21228, USA
*   Correspondence: sbranco@sbu.edu

**Abstract:** Evidence suggests Colombia's transnational adoption program maintained systemic problematic practices, some of which were illicit in nature. Examples include child and birthmother trafficking, sale of children, and falsifying or omitting information in adoption documentation. Transnationally adopted Colombian adults encounter significant barriers to accessing their right to know their origins and identity. Despite this, some adult Colombian adoptees are successful in searching for and engaging in birth family reunions. Our study conducted a secondary analysis of an original study on Colombian birth family reunion experiences. We asked the research question, "What discrepancies exist in Colombian transnational adoption narratives?" to perform a directed qualitative content analysis of 17 participant interviews. We found nearly half of our participants reported an illicit practice categorized as child for sale, birthmother trafficking, and abuse of process. Findings underscore the legacy and impact of harmful adoption practices on current adult Colombian transnational adoptees seeking their human right to identity.

**Keywords:** Colombian adoptees; transnational adoption; illicit adoption practices

## 1. Introduction

Transnational adoption refers to the exportation of children from one country, often defined as developing, to another, industrialized or first-world country (Baden et al. 2015; Molinero and Clemente-Martínez 2021). Frequently, transnational adoptions are also transracial where children, who are members of the Black, Indigenous, and People of Color (BIPOC) communities, are a different race or ethnicity from their adoptive parent(s) who are majority identified as White (Barn 2013; Krieder and Lofquist 2014; Vandivere et al. 2009). Transnational adoption in the United States is widely recognized as starting post-Korean War (Brumble and Kampfe 2011) and spreading to other countries throughout the globe (Lovelock 2000) until the present day. Colombia, the fourth largest nation in South America (Bushnell 1993), was a top exporting nation until it ceased its transnational adoption program in 2013 (Gonzalez 2013). Despite a gradual decline in transnational adoptions around the world within the past two decades (Krieder and Lofquist 2014), the vestiges of the once flourishing practice remain for transnationally adopted adults. One such remnant includes those transnational adult adoptees who seek information about their original identities and birth families. Article 8 of the Convention of the Rights of the Child, a United Nations decree enacted in 1989, delineated how children involved in transnational adoption must have proper procedures in place to maintain the right to know their origins (United Nations 1990). Yet, illicit adoption practices created barriers to ensure all adult transnational adoptees have access to their original identities.

Palacios et al. (2019), in their report on the implications of transnational adoption for the child, noted how unethical practices took place within a system created with minimal oversight or regulation. Specifically, they stated, "Over time, the practice of adoption has been challenged by troubling unacceptable evidence about stolen babies, the oppression

of birth parents and the abuse and/or neglect of children" (Palacios et al. 2019, p. 58). Although various international laws were eventually instituted to provide legal protections and structure for birth families and children, corrupt practices continued to proliferate largely in response to the commodification of children (Molinero and Clemente-Martínez 2021). Multiple scholars described a neo-colonial, transnational adoption industrial complex in which adoption agencies and receiving country authorities were complicit (Briggs 2012; Hübinette 2015; Kawan-Hemler 2022; McCullough 2012; McKee 2016). In Western countries, including the US, the demand for healthy babies and young children outweighed the dwindling supply (Marr et al. 2020). Hence, conditions were ripe for the emergence of corrupt practices to provide adoptive parents children from developing nations (Leinaweaver and Seligmann 2009; Lovelock 2000) and exploited children and mothers in the process (Herrmann and Kasper 1992). Adoption agency professionals profited from illicit practices and the falsification of documents (Associated Press 1998).

Research on the challenges of identity development for transnational adoptees often centers around nonexistent and or lack of birth family information (Baden et al. 2013; Darnell et al. 2017; Koskinen and Böök 2019). Colombian adult adoptees who were adopted in the 1960s through 1990s have experienced difficulties obtaining accurate information about their identities (Branco 2021; Carreazo 2016). Impediments to their right to identity include false or improperly registered birth certificates, falsified, or omitted documentation, and child trafficking (CEVCR 2022; El Tiempo 1986; Emblin 2018; Hoge 1981). Applying the framework of the Right to Identity, the authors aim to shed light on illicit Colombian transnational adoption practices as evidenced through adult adoptees in the United States who have reunited with their Colombian birth families and discovered their adoption narratives revealed problematic practices. Throughout the article, the authors apply the term "birth" to denote the Colombian biological families of adoptees; however, we acknowledge other terms, to include first, natural, or original, may be used interchangeably dependent on preference.

## 2. The Right to Identity and Transnational Colombian Adoptions

Article 8 of the 1989 United Nations Convention of the Rights of the Child (UNCRC) included a child's right to identity. Dambach and Jeannin (2021) described this right as encompassing a "child's name, nationality, and family relations" (p. 4). The UNCRC Article 8 also included provisions for re-establishing a child's identity in the case of illegal withholding or tampering. Article 7 of the UNCRC outlined a child's right to know who and be cared for by their birth parents and charged originating countries with establishing legal structures to enact and monitor rights to identity. The 1993 Hague Convention, created to offer transnational legal safeguards for all parties involved in adoption, also stipulated, in Articles 16 and 30, that actions be taken for sending countries to document and preserve the child's original identity (HCCH 2008).

While the UNCRC and Hague Convention codified the importance of the human right to identity, structural and systemic issues within transnational adoption created circumstances preventing many adult transnational adoptees from retrospectively accessing their identifying birth information (Dambach and Jeannin 2021). The legal process of transnational adoption historically created a permanent cutoff from the child's original birth family and culture and the child was forcibly assimilated to the adoptive parent(s) culture (Baden et al. 2012). Legal remedies implemented in the 1980s and 1990s did little to retroactively support those transnational adoptees adopted years prior. Dambach and Jeannin (2021) recommended accurate information about the child's birth family and circumstances leading up to adoption be documented and provided to the child. They warned the more "gaps" in a child's pre-adoptive history the more harm will be incurred to the child extending into adulthood to develop an identity built on a "solid foundation" (p. 14).

Illicit transnational adoption practices include both illegal and unethical adoption activities (Long 2020). Examples include child trafficking, falsification of documentation,

birth mother coercion, and omission of accurate adoption narrative information (Brown and Roby 2016; Long 2020). In 2020 the Hague Convention convened its Working Group on Preventing and Addressing Illicit Practices in Intercountry Adoption. The group issued a Draft Tool Kit to include fact sheets, checklist, model procedures, and guidelines (HCCH 2020) for adoption agency professionals and receiving country authorities to respond to current and past illicit adoption practices. The Draft Tool Kit was approved in the Working Group's 2022 meeting. Exemplar Tool Kit Fact Sheet recommendations included (but were not limited to) the following topics (HCCH 2020, p. 2):

- Abduction, sale of and traffic of children;
- Misrepresentation of identity;
- Misrepresentation of adoptability of children of unknown parents;
- No preservation of, and access to, records.

Intercountry Adoptive Voices (ICAV) represents a broad group of adult transnational adoptees from around the globe. Their report, submitted to the Hague Convention's Working Group in 2020, highlighted the deleterious impact of illicit transnational adoption practice on adult adoptee's right to identity (Long 2020). A sample of ICAV recommendations included the following:

- International recognition, acknowledgement, and reparation of illicit adoption practices;
- Reparative supports to include post adoption services such as trauma-informed counseling, DNA testing, and central authority support to locate birth family members;
- Legal consequence to those found guilty of engaging in illicit adoption activities.

One specific group of transnational adult adoptees impacted by illicit adoption circumstances include Colombian adoptees adopted during the 1960s through 1990s (Branco 2021; Carreazo 2016; Kawan-Hemler 2022). A confluence of the Colombian National Adoption system created to maintain secrecy and obfuscation of an adoptable child's birth family (Hoelgaard 1998), child trafficking, birth mother coercion for financial gain (El Tiempo 1986; Hoge 1981), the displacement of large numbers of people because of the ongoing civil war (CEVCR 2022), and overall national sentiment that Colombian children would fare better being raised by White westernized families rather than impoverished Colombian families (Hoelgaard 1998; Maestranzi 2013) resulted in 50,000 Colombian transnational adoptions (CEVCR 2022). Adoption documentation was falsified, sealed, nonexistent, and/or destroyed, which created substantial, and in some cases, permanent barriers to accessing rightful identity (Branco 2021; Carreazo 2016; CEVCR 2022). The first author's 2021 case study featured four transnational Colombian adult adoptees raised in the United States who learned, through a birth family search effort, that their adoption records were falsified or otherwise missing information. Such scenarios prevented them from moving further with their search for birth families and created significant impediments to their quest for their original identities. Yet, despite experiencing stigmatization, discrimination, and trauma in their adoptive countries (CEVCR 2022), and then encountering financial requirements, language barriers, and systemic obstructions in their birth family search, some persistent adult Colombian adoptees have been successful in finding their birth families and shared their experiences (Branco et al. 2022). In some cases, their narratives also featured illicit adoption practices which the authors aim to highlight in this secondary analysis.

### 3. Materials and Methods

The data set utilized in the secondary analysis research for this article was derived from an original study entitled, First Family Reunion Experiences of Transnationally Adopted Colombian Adults, of which the first author was lead investigator. For the original study, the researchers conducted semi-structured interviews with 17 adult Colombian adoptee participants who were adopted into the United States. We utilized thematic analysis methodology to manually transcribe, code, categorize and determine themes. Please see Table 1 for demographics of all 17 participants. An emergent theme of the original

research related to inaccurate adoption narratives, illicit, or unethical adoption practices was not enfolded into the research findings as it stood outside of our research questions. In the current study, we conducted a secondary data analysis (SDA) of all 17 participant interviews applying the framework of a new research question, "What discrepancies exist in Colombian transnational adoption narratives?".

**Table 1.** Participant Demographics.

| Participant [1] | Age | Gender |
|---|---|---|
| Sergio | 36 | M |
| Nelly | 42 | F |
| Kate | 28 | F |
| Alberto | 35 | M |
| Sophie | 36 | F |
| Elkin | 29 | M |
| Alex | 23 | M |
| Daniela | 32 | F |
| Elsa | 44 | F |
| Natalia | 37 | F |
| Maria | 40 | F |
| Jennifer | 26 | F |
| Melinda | 34 | F |
| Lina | 30 | F |
| Miguel | 42 | M |
| Fernando | 33 | M |
| Kathryn | 40 | F |

[1] Participants selected their own pseudonyms.

For the SDA, the second author, who has both clinical and research experience with adult transnationally adopted Colombians, joined the lead author, a licensed professional counselor, assistant professor, and transnational adult Colombian adoptee herself, to offer a new perspective to review the original data set. We completed a directed qualitative content analytical (DQCA) approach (Assarroudi et al. 2018; Hseih and Shannon 2005) for the SDA. First, we applied Brown and Roby's (2016) categorization of illicit transnational adoption practices to develop a matrix to include three main categories and two subcategories. The categories are as follows: (1) sale of children, (2) birth mother trafficking, and (3) abuse of process (Brown and Roby 2016). The two subcategories fall under abuse of process and include (1) adoptive parent withholding identifying information and (2) microfiction(s) (Baden 2016). Baden (2016) coined the term microfictions as part of larger terminology surrounding microaggressions in adoption resultant from longstanding stigmatizing attitudes and beliefs. Microfictions are the "mistruths and stories created about adoption that deny or misrepresent, real, lived, adoption experiences" (Baden 2016, p. 6). Please see Table 2 for a complete review of the matrix definitions. Next, we reviewed all the uncoded transcriptions and independently coded passages corresponding to the predetermined matrix definitions. Finally, we compared coding iterations and reached interrater consensus.

**Table 2.** Illicit Transnational Adoption Practice Definitions [1].

| Category | Definition |
| --- | --- |
| Sale of Children | When a child is "transferred for profit or when financial inducements are used to obtain parental consent" [2] |
| Birth Mother Trafficking | Adoption related trafficking where birth mothers are targeted and victimized and can include coercion, fraud, deception, abuse of power, and or forced labor and delivery. [3] |
| Abuse of Process | When an "element of exploitation" [4] or being used unfairly for another's advantage towards the adoptee, adoptive or birth parent, exists in adoption practice. |
| Sub 1: Microfiction | Adoption histories that are "purposefully or accidently" [5] altered, omitted, or inaccurate. |
| Sub 2: AP Withholding | Adoptive parent(s) purposely withholding adoptee identity related information from the adoptee. |

[1] (Brown and Roby 2016). [2] (Brown and Roby 2016, p. 71). [3] (Roby and Brown 2015). [4] (Brown and Roby 2016, p. 72). [5] (Baden 2016, p. 8).

There are limitations with SDA in our study. We were not able to re-interview participants for member checking related to adoption narrative discrepancies, although member checking did occur in the original study. Additionally, because the original research questions were not focused on illicit adoption practices, we found a limited number of participants who shared adoption narrative discrepancies. Although nearly half of the original participants shared an identified false adoption narrative, the sample size is small and cannot be generalized to other adult transnational Colombian adoptees. Finally, the information shared was told to the participant by their birth family, most often their birth mothers, and not shared with us directly. Ultimately, SDA aims to share new information while also lessoning the emotional labor expended by participants in the original study (Ruggiano and Perry 2019). To increase the rigor of the study, we followed SDA best practice guidelines to include (1) clearly describing the original study, (2) securing and sharing Institutional Review Board approval, and (3) detailing limitations (Ruggiano and Perry 2019).

## 4. Findings

Seventeen adult Colombian transnational adoptees who sought out their right to identity via locating and meeting their genealogical birth family relatives shared their reunion stories. Of those participants, eight, or approximately 47%, reported adoption narrative discrepancies discovered during reunion with their birth families. Please see Table 3 for a list of participants as well as reported illicit adoption practice(s) learned during birth family reunion.

**Table 3.** Identified Illicit Adoption Practices.

| Participant | Illicit Adoption Practice(s) |
| --- | --- |
| Alberto | Birthmother trafficking, Abuse of Process: Microfiction [1] |
| Elkin | Sale of Child, Birthmother trafficking, Abuse of Process: Microfiction |
| Sophie | Abuse of Process: Microfiction |
| Daniela | Abuse of Process: AP Withholding |
| Elsa | Abuse of Process: AP Withholding, Microfiction |
| Maria | Abuse of Process: Microfiction |
| Fernando | Abuse of Process: Microfiction |
| Kathryn | Sale of Child, Birthmother trafficking, Abuse of process: Microfiction |

[1] (Baden 2016).

Participants who reported adoption related discrepancies, described the surprise and subsequent feelings such as shock, anger, depression, and overwhelm upon learning of accurate information as told to them from their birth family's perspective. Each participant's examples of illicit adoption practice will be shared next.

### 4.1. Alberto: Abuse of Process–Microfiction, Birthmother Trafficking

Alberto shared the joy experienced upon finding his birth mother. He also described discovering that his adoption story that had been shared with his family by the adoption agency was inaccurate. The abuse of process in the form of a microfiction reflects how Alberto's adoption narrative was falsified or "made up" by their adoption agency. The example is as follows:

> *Yes, so growing up my mom always had my Registro de Nacimiento [registration of birth]. It had my birth mother's name on it but not my birth father. It had my maternal grandparents' names on there. It had their pueblo, where they were from. So, I never knew why I was put up for adoption. And my mom I guess had reached out to the adoption agency, the orphanage [name redacted] to see if they had any more information. I think they just made up a story and just said, 'oh yeah, she was young. She had a boyfriend, and he was like a travel agent and then he just abandoned her, so she gave the baby up for adoption'. It wasn't true.*

Alberto went on to share how his birthmother was coerced into the adoption proceedings at the direction of other family members. We identified the coercion as birthmother trafficking.

> *The real story is that she traveled from (name of town redacted), to (name of city redacted), to live with an aunt who was kind of like her mom, was oldest of her generation. And this aunt, married well and was very educated. And she had other sisters who moved from the pueblo to the (name of city redacted) and they were also very educated and professional. And they hosted my birthmother. And they dressed her up like a maid. And she was kind of like, this, you know, from the countryside and she was supposed to study, but went there to be a maid. And then she had a relationship with one of her cousins and got pregnant. And then, hid her pregnancy. And then, my aunt and my aunt/grandmothers, those professional aunts, were the ones who took her to [name redacted] to put me up for adoption. And so she hadn't spoken to that part of the family since that moment. So, 21 years had gone by.*

### 4.2. Elkin: Birthmother Trafficking, Sale of Child, Abuse of Process–Microfiction

Elkin found his birth family through the support of a private investigator and his own sleuthing based on his adoption papers. He described an emotional reunion and shared a new preadoption narrative recounted by his birth mother. Elkin's story showcases all three areas of illicit adoption practices identified in parenthesis as follows:

> *I was born in '91, the adoption was one of the many (name of state redacted) Colombian adoptees. So my adoption was facilitated by (name of agency redacted). Everything was by the book. You know they had of course all this documentation that this was all consensual, this was all planned out, blah-blah-blah (microfiction). So, it happened 26–27 years later. 25 actually I think, if I'm doing the math right. So, well, anyways, I found her, she immediately asked me if I was ok health-wise. And a bunch of—she—like immediately after her, all of these aunts and uncles and stuff came out of the woodworks— my grandparents. And this is all just still on my maternal side. I still don't know my paternal side. Yea, they all just started coming out of the woodworks and asking me how I am. Am I healthy? Am I ok? And my bio-mom explained that it sounds like I probably was fraudulently taken. Because basically she told me, and my grandparents back up the stories, that they basically approached her like a day after I was born, and they told her, and this is like the Colombian side of things, they told her that I had some rare heart defect, and that I, I needed emergency surgery to live. She was quite poor, and you know they*

*basically just gave her the ultimatum: sign him away, and there's White folks out there with cash in hand or, you know, you can keep him, and he'll die (birthmother trafficking and sale of child). And I was her second child. She had three children after me too that she kept and raised. So she gave me away, and, obviously on our side of things here of the U.S., that's not what shows on paper (microfiction), but, yea, I think they actually attempted to try and find me a few times. They thought, they were pretty convinced that I was either in Germany or England. But I guess they just, you know, I mean on their side of things, once you sign that—the child away, I mean they don't have access to any of that kind of records of where I went. So, and they just did not have the money either, to, you know, really do a good search for me. So, they always hoped that I would find them, and I did.*

### 4.3. Sophie: Abuse of Process–Microfiction

Sophie shared the discovery of an older sister who offered details of her adoption story that were previously unknown to Sophie. We identified this instance as a microfiction, a subtype of abuse of process. Sophie reported, "and then I was able to talk to my older sister, who had a lot more details about my adoption that I had no idea about. And I didn't find her until two years later. She found me". We also identified a subtler, coercive abuse of process related to Sophie's birthmother's employer applying pressure to follow through with relinquishment of Sophie:

*So then, she got pregnant with me. She was cleaning homes. And a woman she cleaned homes for told her you should just give up this baby. Why would you have a fourth, you can't even take care of the three you have.*

### 4.4. Daniela: Abuse of Process–Adoptive Parent Withholding

Daniela described how her search efforts were postponed because of adoptive parent perceived anger about Daniela desiring to find her birth family. Per Daniela, her adoptive mother's sentiment discouraged Daniela from asking for her adoption documentation.

*Yea—I had asked for them, and I was afraid to ask for them, because things had been said when I was younger that kind of just made me—hesitant—like she was—I mean I had known I was adopted since I was really little. And I don't even remember. She doesn't remember saying it, but I remember. I was watching MTV's True Life I'm Adopted, and I just had it on, and that was it. Nothing—I wasn't saying anything, I watched True Life—that wasn't the first episode I watched; it wasn't the last. And my sister—I remember her going into like where my parents' room was, and then my mom coming out and saying, "If you ever look for your birth mother, it'll be a slap in the face" (adoptive parent withholding). So, I think that was part of like—the delay. But then like when all of this happened—like when I say she left it on the table—I literally asked for it, it wasn't a fight, I didn't have to ask for it more than once. It was just kinda—I came home from wherever I'd been, and it was right there. So, I think she was like really supportive. I think she did have some difficulty when they were found, and we were reunited, and I wanted to go see them and all that. I think that was difficult, but I can understand that.*

Further, once Daniela did have her documentation, she encountered delays and challenges accessing more information from Colombian authorities. We classified this also as abuse of process. Access must be simplified and efficient for adoptees seeking their rightful information.

*And then (name of orphanage redacted) put me in touch with someone at ICBF, and then I think they went back to (name of orphanage redacted). Like they needed information. And it was just kind of like a little bit of a process . . . "*

### 4.5. Elsa: Abuse of Process–Adoptive Parent Withholding-Microfiction

Like Daniela, Elsa was also delayed in initiating a search because, unbeknownst to her, her adoptive parents had her birth mother information for years however never

offered them to Elsa. In fact, according to Elsa, her adoptive parents lied about having any knowledge about her birth mother. Hence, we identified two subcategories of abuse of process, adoptive parent withholding and microfiction.

> *Well, this is kind of messed up. My adoptive family had told me that they didn't know anything about her forever (microfiction), and then once I finally was like, set my boundaries and just said, you know, we don't have a relationship anymore. I don't want a relationship with you anymore. They sent me an email. They were like, 'There's some stuff you might want in this [mail]' (adoptive parent withholding). So, I was like, 'Ok, just leave it on your porch.' So, I went there and got it, and her cedula number was on there, and her name was on there. And they had lied to me for 38 years saying that they didn't know anything.*

We also identified abuse of process regarding the apparent poor screening of Elsa's adoptive family as evidenced by Elsa's report of abuse and eventual estrangement. "I'm estranged from my adopted family. It was like a super-abusive, narcissistic, umm, strong addictions kind of family".

### 4.6. Maria: Abuse of Process–Microfiction

Maria described experiencing a period of depression when she learned from her birth siblings that the previously known adoption narrative was in fact false. Maria learned her narrative was not accurate and therefore a microfiction which we define as a type of abuse of process.

> *Then there were some Facebook messages with my adult nieces and nephews, and emails, and then some other information came in that I was not prepared for, like it was different information from what I had heard and so, then I took another break, because it was too much to deal with. Maybe this is helpful. I remember this quite vividly, because they were saying, 'No, that's not true what happened, it was this,' (microfiction) and so I actually—I had—I went into a slight depression for two weeks. And that was a little scary, because that was my first time, and I was a social worker. I was like, I should know what's happening! And it wasn't until my therapist told me, 'You're having some symptoms of depression.' I'm like, 'I am?' So, that actually helped, to name it, and then I was able to come out of it,*

### 4.7. Kathryn: Sale of Child, Birth Mother Trafficking, Abuse of Process–Microfiction

Kathryn recounted her reunion story being filled with new information, some of which was challenging to learn. Kathryn first described how her birth mother was coerced and deceived in the adoption proceedings on more than one occasion. She then described subsequent inaccuracies and errors in her Colombian adoption documentation. We found all three types of illicit adoption practices in Kathryn's story. Examples of birthmother trafficking include the following:

> *Because of the nature of my adoption, it was a complete shock to my birth mother [sic-being found]. And she ended up telling her story when I met her. She had explained that I was conceived when she was raped by a man working at her job, and that her supervisor turned her in to the clinic managed by my orphanage who made her sign lots of papers despite being unable to read and write. She said that she was told I was born dead. So she didn't know all this time that I was alive.*

> *She had a similar experience with her job when she had her next baby after me. She shared that she was drugged during the birthing which allowed the baby to be taken from her without her consent.*

We identified a sale of children example as follows:

> *And I wonder if other people, you know—I'm really like—I really wonder if other people in reunion at this level of, being part of black-market adoption are faced with this, 'cause I don't know of any—I haven't spoken to any other adoptees really—who are experiencing*

*this level of, struggle and challenge. It's not easy to live with finding out that a whole government and a whole Catholic institution, and individuals of a workspace conspired to steal a baby and get away with it. And a challenge for me there, and I'm not saying this for everyone, I'm saying specifically for me it's like I don't know how? I can't sit with this information. Yea, I want justice. Personally. And how do I, one person, take on an entire government of—I mean it's infuriating, and I don't know if it's still happening? Probably sometimes still is? But to argue something that happened in the past is . . . .yea. So, this—so the challenge for me is like this, this, this is a burden on my shoulders now. It's a new burden.*

In addition to absorbing the initial information about her adoption narrative, Kathryn later learned that her adoption documentation was inaccurate hence constituting abuses of process, microfiction subtypes. First, she discovered she had a sibling, and her documentation was missing information.

*Before learning this, my adopting orphanage shared with me that my file showed that I had a sibling also adopted somewhere in the United States or Europe. They wouldn't tell me more than that. But when I received my file eventually through the welfare system, I was infuriated to see two pages were missing. And the pages were so obviously renumbered without any explanation (abuse of process-microfiction).*

Next, the inaccurately filed documentation created a four-year journey to rectifying the errors so that she could obtain her Colombian national identification number, known as a cedula in the Spanish language. Colombian citizenship is the right of all persons born in Colombia and therefore applies to Colombian adoptees. The following example highlights both abuse of process as well as sale of child as noted in Kathryn's description of being a "stolen child".

*Obtaining a cedula takes many steps and I discovered my birth certificate and US passport have different names and therefore I needed to do a legalized name change in Colombia before obtaining my Cedula. Again a 4-year process for me. Then I found out that the welfare system's process is long, antiquated, and filled with inconsistencies and unanswered emails for months. Then I was lucky to receive my file, but pages were missing and renumbered. So there is the avenue of going through the welfare system. There is the avenue of going through your orphanage, or through your, you know, agency. There's the avenue of going through, umm, hiring private help, umm, and I really think that's another good idea if you have the resource, but I don't support the enormous fees that take advantage of the situation. I don't know how many people go through life like literally having to go through so much on their identity through consulates and legal offices across oceans and languages. And so eventually I found out because I was adopted at 30 days old, so many multiple legal adoption steps were skipped, and now I understand this is because it was a stolen situation. I was a stolen baby. There were other steps that were skipped.*

We identified various types of abuse of processes in all eight participants who described illicit practices in their narratives. We found three participants with instances of birth mother trafficking and two participants with sale of children reports. Further we noted participant comments suggestive of problematic practices yet did not fully meet the criteria outlined. For example, Sophie learned that after she had been relinquished for adoption her uncles attempted to adopt her to no avail. "So, when I was given up for adoption, the uncles wanted to adopt me, so they tried to fight it, but it was already done". Current child welfare practices support kinship or family adoption, yet this was not considered at the time for Colombian adoptions. Melinda similarly described how her birthmother worked to rescind the relinquishment with no success:

*So, she was just explaining everything and all the hardships and everything like that, and actually she gave me up, and, within a couple of weeks, she found a job at a, I think at a farm, and she came back, but I had already been adopted out. And that was kind*

*of—I think it was just kind of final for her, and I don't—she never made it seem that she thought that she could find me again.*

Current best adoption practices mandate a period when birth parents may rescind their voluntary relinquishment.

Illicit adoption cases were prevalent in Colombia, causing distress to the adult adoptee when trying to connect to their roots and find out about their story. Participants in our research who found contradicting information in their adoption process illuminate the illegal practices and injustices of the Colombian government. In some of these cases, one could argue that it is the continuation of indigenous genocide, comparable to the US federally funded Indian Adoption Project where Native American children were removed from tribal reservations to be adopted and assimilated into White families (Thibeault and Spencer 2019). Kathryn reports:

*A huge piece of this has been my ongoing work connecting to my indigenous heritage. Since learning that my adoption was non-consensual, it has strengthened my belief that my adoption is part of the continued genocide of indigenous people through forced family separation.*

The Hague convention delineated policies and regulations where family preservation is one of the main goals before considering international adoption; this includes extended family. The sending country must exhaust every option for the child to be adopted within the same country (HCCH 2007). If this is not possible, then international adoption may become an option. These regulations have been established to prevent child sales, abduction, and child trafficking (Rotabi 2013).

Birth mother trafficking, sale of children, and abuse of process were the three main categories found in study's SDA. The consequences of these practices on adoptees are yet to be fully researched. This exploratory study highlights the potential for long-standing wounds in many adoptees and their families as a result of unethical and illicit practices. Institutional attempts to regulate the process of international adoptions have been thwarted by corruption within impoverished sending countries (Rotabi 2013). For example, one participant in the study reported their Colombian birth mothers signed the paperwork relinquishing their parental rights without knowing how to write or read.

## 5. Conclusions

Our findings are corroborated by what we know about past Colombian transnational illicit adoption practice to include corruption, child and birthmother trafficking, the sale of children, and inaccurate or falsified adoption documents and birth registrations (Branco 2021; Committee Investigating Intercountry Adoption 2021; Carreazo 2016; El Tiempo 1986; Hoge 1981; and CEVCR 2022). Nearly 50% of the participants in the original study about Colombian birth family reunions, reported some type of illicit practice in their adoption narratives. The findings are impactful, as uncovering this information was not the aim of the original study. As such, it is imperative that the findings add to the growing body of evidence of illicit past practices and require further investigation. Given the sample size, findings are not generalizable to other transnational Colombian adoptees specifically or transnational adoptees in general. However, information gleaned from the study sheds light on possible similar circumstances for other transnational adoptees from countries with questionable adoption practices to include Chile (Salvo Agoglia and Monsalve 2019), Ethiopia (Branigan 2018), and Guatemala (Siegal 2011). Future research may directly inquire about illicit practices among transnational adult Colombian adoptees as well as Colombian birth families. Most importantly and urgently will be follow-up action undertaken by Colombian government officials to investigate past illicit practices and develop reparative strategies to assist the thousands of adult Colombian adoptees to obtain their human right to identity.

**Author Contributions:** Conceptualization, S.F.B.; methodology, S.F.B.; formal analysis, S.F.B. and V.C.; resources, S.F.B. and V.C.; data curation, S.F.B.; writing—original draft preparation, S.F.B.;

writing—review and editing, S.F.B. and V.C. All authors have read and agreed to the published version of the manuscript.

**Funding:** This research received no external funding.

**Institutional Review Board Statement:** The study was approved by the Institutional Review Board at St. Bonaventure University #477 on 25 April 2022.

**Informed Consent Statement:** Informed consent was obtained from all subjects involved in the study.

**Data Availability Statement:** Data sharing not available due to restrictions from the Institutional Review Board of St. Bonaventure University.

**Conflicts of Interest:** The authors declare no conflict of interest.

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
