# Peer review of "False Narratives: Illicit Practices in Colombian Transnational Adoption"

_genealogy, doi:10.3390/genealogy6040080_

Round 1

Reviewer 1 Report

It is a very important read that contributes to the post colonial adoption theory. It donors the voices of adoptees and their lived experiences that should inform practice.

The article gives an in-depth apperçu of the narratives of the transnational adoptees from Columbia. The narratives reflect on the complex experiences they have been exposed to, especially in terms of their possible efforts to search for their origin and biological family and mother. The narratives also reflect on the processes leading to their transnational adoptions. These narratives intersect with the postcolonial discourse on adoption that is less present in academic writing.The narratives did not reflect on complexities related to identity formation that are the most present  challenges faced by transracial and international adoptees.
In terms of literature review, the article misses any connection with the adoption of Indigenous children in Columbia and other Indigenous territories such Canada and the US, in non-Indigenous families and it also misses any reflection on the worldwide transracial/transnational adoption trends and the role of adoption agencies. It also misses tapping on local legislations vis-a-vis adoption except for the reference to the Hague convention.
Despite this, the article presents rich narratives that might disturb the normative way of perceiving adoption as the best practice in offering alternative family care for children.
The article would benefit from a minor editing exercise.

Reviewer 2 Report

This original and important article is based around a secondary analysis of data from a study on Colombian international adoptees. It examines and discusses illicit adoption practices in adoptees' narratives. It finds that half of the participants in the original study identify their adoptions as having illicit elements, and reveals the broader issues of corrupt, unethical and illegal adoption practices. Such practices occur in the process of the adoption itself (for instance through "birthmother" trafficking, sale of children and falsification of documents), in the adoptee's life (through concealing of background information and being subjected to background micro-fictions), and in the family search process (through withholding of information).

I believe that this work is an important addition to a growing body of critical adoption research. In particular the definitions and examples of illicit adoption practice that emerge through the study will be of great use in future research, as well as the findings. 

The research is methodologically sound, and using SDA is highly appropriate in making the research non-intrusive. The analysis and discussion is clear and interesting to general readers as well as those in the field.

I have a few minor questions, suggestions and comments:

1) The receiving country/ies of the adoptees were not immediately clear to me (USA?). Could the authors please clarify?

2) "birth mother" is a term that many mothers who have lost their children to adoption find offensive. I understand that the authors use this term for clarity, but wondered if "first mother" or similar could be more appropriate? If not, could the authors indicate in the article that they are aware of the problematic nature of the term?

3) There was very little reflection on the role of the adoption agencies and receiving countries' authorities in the discussions of illicit practices. Would it be possible to comment on these actors in the adoption market in some way?

4) The definitions of illicit practices are really significant, and I believe they will not just be useful for other researchers, but also for adoptees trying to understand their own experiences. As such, could the authors expand a little on these, and maybe provide more examples? At the moment they are briefly outlined in Table 2. I. While the short definitions are clear, concepts such as microfictions, could really do with being explained further. 

5) I was confused by the sentence on line 46-line 47 - should the "developed" be "developing"?

6) In the conclusion could the authors connect their research on Colombian adoptees to the bigger picture of international adoption? I am sure their findings would reflect practices relating to adoptions form other sending nations too. 

All in all, I find this a well-written, scientifically sound article, that is original and highly significant.
